# Relationship between Parental Involvement and Mathematics Achievement of Chinese Early Adolescents: Multiple Mediating Roles of Mental Health and Mathematics Self-Efficacy

**DOI:** 10.3390/ijerph18189565

**Published:** 2021-09-10

**Authors:** Feifei Huang, Zhaofeng Huang, Zhe Li, Minqiang Zhang

**Affiliations:** 1School of Psychology, South China Normal University, Guangzhou 510631, China; 2016010162@m.scnu.edu.cn (F.H.); muzilizheli@163.com (Z.L.); 2School of Life Sciences, Guangzhou University, Guangzhou 510006, China; huangzhaofeng@gzhu.edu.cn

**Keywords:** parental involvement, mathematics achievement, mental health, mathematics self-efficacy

## Abstract

This study conceptualized the multidimensional construct of parental involvement, including cognitive involvement, behavioral involvement, and personal involvement, and examined the mediating effects of student’s mental health and mathematics self-efficacy. Questionnaires were administered to 2866 early adolescents and their parents in China; structural equation modeling and bias-corrected bootstrap methods were used. The results show that different dimensions of parental involvement had different effects on mathematics achievement. Additionally, results indicate that the influences of the multidimensional construct of parental involvement on mathematics achievement were either partially or completely mediated by student’s mental health and mathematics self-efficacy. The findings also offer insight into possible interventions designed to explore how parental involvement promotes students’ mathematics achievement through their children’s mental health and mathematics self-efficacy.

## 1. Introduction

Enhancing the educational success of all students is a vital educational goal, one which has received considerable public attention. Therefore, it is critical to identify impact factors that promote academic success and the pathways through which they operate [1]. One promising avenue for improving students’ academic outcomes is parental involvement [2]. Parents play a vital role in their children’s education [3]. Especially, studies have shown that children have higher achievement if their parents are more involved in their education [4,5]. It is well known that educating children is a very complex process that demands the collaborative efforts of both parents and schools [6,7]. Although a substantial body of literature has highlighted the important influence of parental involvement on general academic performance [8,9,10], much less work has focused attention on the links between subject-specific measures of achievement and parental involvement. It could be possible that students’ have better achievement in subjects that their parents care about more. Studies have indicated that the knowledge of mathematics, as a strong foundation for student’s future academic performance, plays an important part in the educational development of youths [11,12,13,14,15]. Additionally, it is well known that mathematics can be regarded as an indispensable skill in all facets of life. Further, mathematics achievement can be considered to be the vital factor that contributes to the excellence of the education sector [16]. Thus, it is necessary to explore the domain-specific relations between students’ mathematics performance and parental involvement.

Researchers have conceptualized parental involvement as a multifaceted construct, including cognitive/intellectual involvement, behavioral involvement, and personal involvement [15,17]. The association between academic performance and parental involvement appears to vary by types of involvement [18,19]. However, little research has investigated how and to what extent each dimension of parental involvement is related to mathematics achievement [20]. In addition, it is unknown how the underlying mechanisms of parental involvement affect students’ mathematics achievement. Previous research has identified that both students’ mental health and self-efficacy were strong predictors of academic achievement [21,22]. These two key factors are malleable and related characteristics, making a great difference on students’ achievement. Taken together, the multidimensional construct of parental involvement may play an important role in enhancing children’s mathematics achievement through its impact on students’ mental health and mathematics self-efficacy.

### 1.1. Parental Involvement and Mathematics Achievement in Elementary School

In recent years, parental involvement has been a new focus of public concern. Researchers have conceptualized parental involvement as “parents” interaction with schools and with their children to promote children’s academic success [23]. Researchers have incorporated the multidimensional construct of parental involvement by identifying specific components [2,17,18,24]. Grolnick and Slowiaczek (1994) proposed a three-dimensional model for exploring particular parents’ influence on their children’s educational outcomes, including cognitive/intellectual involvement, behavioral involvement, and personal involvement [17,19,25,26]. First, cognitive/intellectual involvement refers to whether the children are exposed to educationally stimulating activities and experiences. Second, behavioral involvement, refers to parental involvement through their by their participatory behavior in activities, such as assisting children with homework or volunteering at school. Finally, personal involvement, refers to parental socialization around the attitudes and expectations about school and education [17,19].

Studies have found that the three dimensions of parental involvement have an impact on students’ educational success [19,25,26]. However, different dimensions of parental involvement yield different effects on students’ achievement [17,26]. Behavioral involvement and cognitive involvement may have a stronger effect on student achievement. These kinds of parental involvement may have a positive influence on children’s learning process [17,26]. In contrast, personal involvement seems to be most weakly related to student achievement. Parents may focus more on sharing the affective experience of caring about school than academic achievement [17,26]. Further, as mentioned before, the knowledge of mathematics, which is a strong foundation for a student’s future academic performance, plays an important part in a youth’s educational development [11,12,13,14,27]. Therefore, it is advisable to consider the relationship between the various facets of parental involvement and mathematics achievement.

### 1.2. Mental Health as a Mediator between Mathematics Achievement and Parental Involvement

Mental health, a state of well-being, is regarded as a positive force that promotes the healthy development of individuals [28]. Although people come to realize the importance of mental health, the incidence of mental health disorders among children and youth is still increasing, such as depression, loneliness, anxiety, and so on [1,29].

To raise the awareness of mental health, numerous studies have found some factors that may have impact on students’ mental health. Recent studies suggest that higher parental involvement contributes to raising the level of children’s psychological health [22]. In essence, parental involvement could meet the psychological needs of students, which in turn promotes positive mathematics achievement.

Students’ mental health seems to be a reasonable mediator in the relation between parental involvement and mathematics achievement. For one thing, the school–parent relationship, a critical factor for young people’s development, is considered to have a substantial effect on children’s mental health [1,30,31,32]. In fact, studies have shown that the lower odds of students’ poor mental health were significantly associated with high levels of self-reported parental involvement [33,34,35]. Additionally, a growing body of literature has demonstrated that both students’ psychological needs and current priority in education have a positive effect on students’ academic achievement [36,37,38,39]. Similar results were reported by Sherman and Wither (2003), who emphasized the importance of mental health as a critical feature of math-related achievement [40]. It is not surprising that children who possess a higher level of mental health may have higher academic achievement in mathematics. Hence, students’ mental health may serve as a plausible mediator in the relationship between mathematics achievement and parental involvement.

### 1.3. Self-Efficacy as a Mediator between Mathematics Achievement and Parental Involvement

Self-efficacy, a pivotal component of social cognitive theory [41], is defined as an individual’s capabilities to perform or achieve an expressed goal [42,43]. In addition, mathematics self-efficacy is the belief in one’s own ability to succeed in mathematics, which is significantly and positively correlated with mathematics performance [44,45]. One study has suggested that people who have higher self-efficacy beliefs tend to act more confidently and persistently in the face of obstacles [46]. Moreover, the relation between mathematics achievement and mathematics self-efficacy is of great interest to educators and researchers [21].

The role that parental involvement may play in influencing children’s mathematics achievement indirectly through mathematics self-efficacy is theoretically compelling. For one thing, some researchers have demonstrated that the impact of parental involvement on self-efficacy is of great importance in real life [47]. In a similar vein, Fan and Williams’s (2010) study has also confirmed a pattern in which parental involvement positively influences mathematics self-confidence [48]. Moreover, studies have found that mathematics self-efficacy, a domain-specific self-efficacy evaluation, has been regarded as a well-established predictor of mathematics achievement indexes [49,50]. Similarly, Levpušcek’s (2013) study also indicated that mathematics self-efficacy positively and significantly correlates with children’s mathematics learning and problem solving [26,51]. As such, mathematics self-efficacy may play a mediating role in the relationship between parental involvement and mathematics achievement. Moreover, there is evidence to suggest that math-related self-efficacy is posited as a mediating construct that connects parental involvement and mathematics for students [52]. Nonetheless, few studies have explored different aspects of parental involvement as a more robust predictor of a student’s sense of mathematics self-efficacy. Thus, the present study examined whether different types of parental involvement exert a positive impact on children’s mathematics performance via the mediating effect of mathematics self-efficacy.

### 1.4. Role of Covariates

Although this study focused on examining the relationship between different types of parental involvement and mathematics achievement, other background variables such as students’ gender, numbers of children at home, and a family’s socioeconomic status (SES) should also be considered. Studies have indicated that each of these variables has been substantially associated with both parental involvement and mathematics achievement [1,9,20].

### 1.5. The Current Study

The aim of this study was to extend previous research by examining the direct effects of different dimensions of parental involvement on students’ mathematics achievement and to consider the multiple mediating roles of students’ mathematics self-efficacy and mental health in the parental involvement and mathematics achievement (see Figure 1). Specifically, we sought to identify the extent to which the dimensions of parental involvement promoted student’s mathematics achievement and to understand whether parental involvement influenced students’ mathematics achievement by increasing students’ mathematics self-efficacy and mental health. In short, this study not only aimed to identify the particular parental involvement strategies that contribute to mathematics achievement but also aimed to explore the effects of the multifaceted construct of parental involvement and students’ mental health and mathematics self-efficacy on mathematics achievement from a multiple mediating perspective.

Hereupon, based on the literature review above, we formulated two hypotheses:

(1) Parental involvement was positively associated with mathematics achievement. Different dimensions of parental involvement would be strongly related to mathematics achievement.

(2) Parental involvement would promote early adolescents’ mathematics achievement by enhancing their mental health and mathematics self-efficacy, based on the multiple mediation model.

## 2. Methods

### 2.1. Participants and Procedures

The data analyzed in the present study were from the Comprehensive Assessment of Basic Education Project, conducted by Teaching and Research Department in China. Its main purpose was to test the development of students’ achievement on specified subjects and its correlated influencing factors on students’ learning. To this end, a standardized test in math as well as questionnaires for students, parents, teachers, and school principals were administered. A total of 144 students chose not to be included. Therefore, the final data contained 2866 (53.24% boys) sixth graders and their parents in a city located in south China. The early adolescent participants ranged in age from 8 to 13 (Mean (*M*) = 11.57, Standard deviation (*SD*) = 0.57).

Before the study was conducted, ethical approval was obtained from the local Ethics Committee of the School of Psychology, South China Normal University. All participants provided written informed consent. Researchers clarified the purpose of the study to the participants and assured the participants of the confidential and voluntary nature of the study. Both the math test and questionnaires were administered to all students in their classrooms during school hours. Data from students’ parents were collected via online questionnaires.

### 2.2. Measures

#### 2.2.1. Parental Involvement

Consistent with Grolnick and Slowiaczek (1994), three facets of parental involvement were measured: cognitive, behavioral, and personal involvement [17]. In each case, six items were used to assess a parent’s perception of the positive attitude toward their child’s school education. Part of the items listed as follows: “Collect learning resources for children through the Internet” (cognitive involvement); “Setting up relevant study plans, or arranging daily schedules for your children” (behavioral involvement); “In depth communication with your children after a short time” (personal involvement). For each item, parents were required to indicate on five-point Likert scale (1 = strongly disagree, 5 = strongly agree) how often they had behaved similarly. Scores for the three aspects of parental involvement were computed separately, with higher scores representing higher involvement. Confirmatory factor analyses indicated that a three-factor model fit the data satisfactorily for parental involvement: *χ^2^/df* = 12.10, comparative fit index (*CFI*) = 0.93, Tucker and Lewis index (*TLI*) = 0.92, root-mean-square error of approximation (*RMSEA*) = 0.06, standardized root-mean-square error (*SRMR*) = 0.04. In the current sample, the three aspects of parental involvement had Cronbach’s *α* of 0.82, 0.84, and 0.83 for cognitive involvement, behavioral involvement, and personal involvement, respectively. Additionally, the measures had a composite reliability of 0.82, 0.85, and 0.84 for cognitive involvement, behavioral involvement, and personal involvement respectively.

#### 2.2.2. Mental Health

Adapted from an existing instrument [53], six items measuring mental health were used in the context of Chinese culture. Students’ response to these items reflected the level of students’ mental health. Each item was rated on five-point scale from (1) “strongly disagree” to (5) “strongly agree”. Sample items include, for example, “I can deal well with unhappy feelings” and “I can always find a way to make myself feel better when I upset”. Confirmatory factor analyses indicated that the model fit the data satisfactorily: *χ^2^/df* = 21.47, comparative fit index (*CFI*) = 0.97, Tucker and Lewis index (*TLI*) = 0.93, root-mean-square error of approximation (*RMSEA*) = 0.08, standardized root-mean-square error (*SRMR*) = 0.03. The instrument had a Cronbach’s *α* of 0.80 in the current sample. Additionally, it had a composite reliability of 0.80 in the current sample.

#### 2.2.3. Mathematics Self-Efficacy

This was assessed by using the mathematics self-efficacy questionnaire from the short form of the Mathematics Self-Efficacy Scale [54]. The Mathematics Self-Efficacy Scale is a 7-item self-report measure designed to assess student’s confidence about mathematics study. Respondents endorsed each item on a 4-point Likert-type scale ranging from 1 (strongly disagree) to 4 (strongly agree). Confirmatory factor analyses indicated that the model fit the data satisfactorily: *χ^2^/df* = 15.82, comparative fit index (*CFI*) = 0.97, Tucker and Lewis index (*TLI*) = 0.95, root-mean-square error of approximation (*RMSEA*) = 0.07, standardized root-mean-square error (*SRMR*) = 0.03. It had a Cronbach’s *α* of 0.81 in the current sample. In addition, it had a composite reliability of 0.81 in the current sample.

#### 2.2.4. Mathematics Achievement

The students’ mathematics score from a standardized test was used as the index of mathematics achievement, which ranged from 0 to 100.

#### 2.2.5. Covariates

Students’ gender (0 = male, 1 = female), number of children at home (0 = only one child, 1 = more than one child), and SES were used as covariates in the models. SES was measured using a composite of parental education (0 = no college degree, 1 = college degree) and annual family income (1 = ¥12,000 or less, 2 = ¥12,001–¥24,000, 3 = ¥24,001–¥36,000, 4 = ¥36,001–¥48,000, 5 = ¥48,001–¥60,000, 6 = ¥60,001–¥72,000, 7 = ¥72,001–¥120,000, 8 = over ¥120,000). We standardized and averaged both indicators to create the SES composite score [1].

### 2.3. Analytic Strategy

First, we used descriptive statistics and correlation analyses to examine the association between parental involvement, mental health, and self-efficacy and mathematics achievement. Second, independent-samples *t*-tests were conducted to examine whether there were any statistically significant associations between the demographic variables (gender, number of children at home, SES) and the main variables. Third, structural equation modeling (SEM) was performed to test the hypothesized relations among the study constructs. The assessment of the SEM model was based on the fit index of a chi-square test, the comparative fit index (*CFI*), the Tucker–Lewis index (*TLI*), the root-mean-square error of approximation (*RMSEA*), and the standardized root-mean-square error (*SRMR*). The model was acceptable when the fit indices met the following criteria: *χ^2^/df* < 3, *TLI* and *CFI* > 0.90, *RMSEA* < 0.08, *SRMR* < 0.08. Finally, a bias-corrected bootstrap method with 1000 bootstrap samples was used to confirm the significance of mediation effects by 95% confidence intervals [55]. SEM was conducted using Mplus 7.0 software (Muthén & Muthén, Los Angeles, CA, USA, Available at: http://www.statmodel.com/) [56], and we used SPSS 19.0 (IBM SPSS Statistics, Armonk, NY, USA) to conduct other data analyses.

## 3. Results

### 3.1. Preliminary Analyses

The means, standard deviations, and bivariate correlations of the study variables are presented in Table 1. The means for each dimension of parental involvement indicated that parents were relatively more involved in their behavioral involvement (*M* = 4.22), personal involvement (*M* = 4.28), and cognitive involvement (*M* = 4.15). Moreover, preliminary analyses were conducted to examine the effects of the number of children at home, gender, and SES on the key variables. An analysis of variance (independent samples test) revealed that children whose family had only one child performed better on parental involvement, mental health, mathematics self-efficacy, and mathematics performance than those with two or more children. No significant difference was found for gender on any of the key variables. Higher SES parents reported more parental involvement than lower SES parents.

### 3.2. Tests of Measurement Model

Before testing our structural model, we tested the measurement model as a preliminary step. The fit indices are presented in Table 2. The results of confirmatory factor analysis (CFA) show that the hypothesized model fit the data sufficiently, except for the *χ^2^* value being significant because of the sensitivity of the sample size. In addition, all corresponding factor loadings for each factor were statistically significant at the 0.05 level, with moderate to high magnitudes.

### 3.3. Structural Equation Model

The structural equation model was assessed after the measurement model was accepted. To test multiple mediation effects, a multiple mediation model was used to better understand the separate contribution of each individual mediator and to examine whether those potential mediators jointly reduced the direct effect of parental involvement on a students’ mathematics achievement [57].

#### 3.3.1. Direct Effects of Parental Involvement on Mathematics Achievement

We examined the direct association between mathematics achievement and parental involvement (see Figure 2). The overall model fit was good, *χ^2^/df* = 18.12, *RMSEA* = 0.07 (0.066, 0.075), *CFI* = 0.90, *TLI =* 0.87, and *SRMR* = 0.06. Cognitive involvement and behavioral involvement were positively associated with mathematics achievement (*β* = 0.16 and 0.09, respectively), whereas personal involvement was not associated with mathematics achievement directly.

#### 3.3.2. The Multiple Mediation Effects

The multiple mediation analyses examined the extent to which students’ mental health and mathematics self-efficacy mediated the associations between different types of mathematics achievement and parental involvement (see Figure 3). The multiple mediation model fit the data reasonably well (*χ^2^*/*df* = 9.56, *CFI* = 0.94, *TLI* = 0.92, *RMSEA* = 0.06 (0.058, 0.061), *SRMR* = 0.06). Table 3 shows the results of direct, indirect, and total effects in the final model. Three types of parental involvement (cognitive, behavioral, personal involvement) were positively associated with students’ mathematics self-efficacy (*β* = 0.16, 0.09, 0.07, respectively), whereas only cognitive involvement and behavioral involvement were positively associated with students’ mental health (*β* = 0.17 and 0.13, respectively). Moreover, the association between cognitive involvement, behavioral involvement, and mathematics achievement was partially mediated by students’ mental health and mathematics self-efficacy. The association between personal involvement and mathematics achievement was fully mediated by mathematics self-efficacy. In general, cognitive involvement and behavioral involvement influenced mathematics achievement directly and also indirectly through both students’ mental health and mathematics self-efficacy, whereas personal involvement influenced mathematics achievement only through mathematics self-efficacy indirectly.

## 4. Discussion

In the present study we conceptualized parental involvement as a multidimensional construct and examined the relationship of its dimensions with school performance. Of particular interest to the present study is the result that focuses on mathematics achievement in elementary school. Additionally, this study contributes to literature by achieving two major goals. The first was to investigate the relationship between different dimensions of parental involvement and mathematics achievement, presented in Section 4.1. In addition, the second goal was to evaluate the multiple mediation mechanisms through which these different types operate, presented in Section 4.2. Then, the implications of this study are presented in Section 4.3. Finally, we discuss some limitations at the end of Section 4.3 and present the general conclusions of this study in Section 5.

### 4.1. Parental Involvement and Mathematics Achievement

With regard to the first goal, the results indicate the necessity of distinguishing among different types of involvement in which parents engage. The results are in agreement with the previous studies in finding that cognitive involvement and behavioral involvement were proved to have a significant and direct impact on mathematics achievement [17,19,26]. In other words, parents who are more involved in their children’s school performance may also place greater importance on their children’s school progress, which contributes to improving their child’s mathematics achievement. Studies have found that, compared with the Western countries, there is a strong academic-centered orientation in Chinese schools [58,59,60,61,62], which means students in China with better scores are normally favored by their peers. Therefore, children’s academic performance may facilitate parental practices with respect to children’s learning [3]. However, personal involvement was not statistically significantly or directly related to mathematics achievement. The possible explanation of this is that parents were more concerned with their children’s emotions, neglecting the academic issues.

### 4.2. The Multiple Mediation Role of Mental Health and Mathematics Self-Efficacy

The second goal of the study was to examine a multiple mediation model in which student’s mental health and mathematical self-efficacy are mediators between parental involvement and mathematics achievement. The results were generally consistent with the hypothesized model. There were indirect associations between different dimensions of parental involvement and mathematics achievement through the mediators, yet these pathways were only in evidence for some dimensions of parental involvement.

Of importance, our findings show that two dimensions of parental involvement—both cognitive involvement and behavioral involvement—played a partly mediating role. That is to say, the association between these two factors of parental involvement and mathematics achievement were mediated by student’s mental health and mathematics self-efficacy. There is a possibility that parents who engage in school or attend cognitive activities may contribute to children’s mental health [1,17]. Moreover, these behaviors may result in children’s good performance on tests of mathematics achievement. The feedback loop suggests a cycle in which parental involvement maintains the ongoing performance of children. Notably, although personal involvement appears to have no direct effects on mathematics achievement, our findings still show that it can contribute to mathematics achievement indirectly through mathematics self-efficacy, which means personal involvement plays a fully mediating role. It is possible that such emotional closeness with parents helps the children to construct a sense of positive self-efficacy.

The relations between parental involvement and mathematics achievement were examined when controlling for the number of children at home, students’ gender, and SES. Our results were consistent with the previous literature showing that the number of children at home was significant in predicting parental involvement on mathematics achievement [63]. Parents who have only one child tend to be more involved in school activities. Additionally, we found that gender does not have a statistically significant impact on the associations between parental involvement and mathematics achievement. It is possible that relevant measures to improve gender equity have begun to work. In light of the current findings, this study also showed that SES was significant in determining the level of parental involvement and mathematics achievement. Higher SES parents were more involved in their children’s education. There is the possibility that lower SES families may have limited resources [64].

### 4.3. Educational Implications

The present study adds to the volume of literature by indicating that parental involvement predicts mathematics achievement in elementary school students via the student’s mental health and mathematics self-efficacy. The path—parental involvement→students’ mental health and mathematics self-efficacy→students’ mathematics achievement—illuminated the multiple mediating roles in the relations between parental involvement and mathematics achievement. The results indicate that different dimensions of parental involvement had different effects on mathematics achievement. The influence of cognitive involvement and behavioral involvement on mathematics achievement was partially mediated by students’ mental health and mathematics self-efficacy, whereas the relation between personal involvement and mathematics achievement was fully mediated by the above factor. These results further support the viewpoint that cognitive, behavioral, and personal involvements are unique constructs. In addition, these findings also have implications for parents on how they can play a pivotal role in their children’s mathematics achievement. Specifically, parents can enhance their children’s mathematics achievement by providing them with more involvement. In this way, students will develop higher mathematics self-efficacy and a higher level of mental health. Therefore, increasing the likelihood that such students can achieve higher scores on mathematics.

The conclusions drawn from the current study should be considered in light of a number of limitations. On one hand, the sample was drawn from a city of China; thus, participants in our study may not be representative of the other populations. On the other hand, there may be additional mechanisms underlying the associations. Further exploration into underlying mechanisms could promote interventions in support of students’ mathematics achievement. Despite these limitations, the current study makes a contribution to the relevant literature between different dimensions of parental involvement and mathematics achievement while incorporating a multiple mediation model to examine separate mechanisms. Future research aiming to promote mathematics achievement might focus on the specific roles of students’ mental health and mathematics self-efficacy, not just parental involvement.

## 5. Conclusions

This study conceptualized the multidimensional construct of parental involvement, including cognitive involvement, behavioral involvement, and personal involvement, and explored the effects of the multifaceted construct of parental involvement and students’ mental health and mathematics self-efficacy on mathematics achievement from a multiple mediating perspective. The results show that different dimensions of parental involvement had different effects on mathematics achievement. In addition, the results indicate that the influences of the multidimensional construct of parental involvement on mathematics achievement were either partially or completely mediated by students’ mental health and mathematics self-efficacy. The findings also offer insight into possible interventions designed to explore how parental involvement promotes students’ mathematics achievement through their children’s mental health and mathematics self-efficacy.

## Figures and Tables

**Figure 1 ijerph-18-09565-f001:**
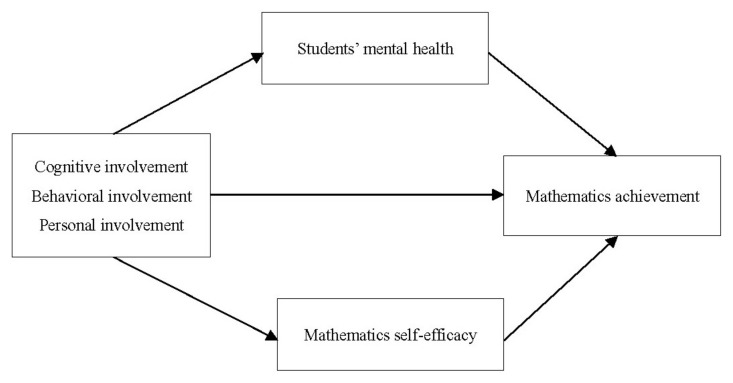
Hypothesized multiple mediator model.

**Figure 2 ijerph-18-09565-f002:**
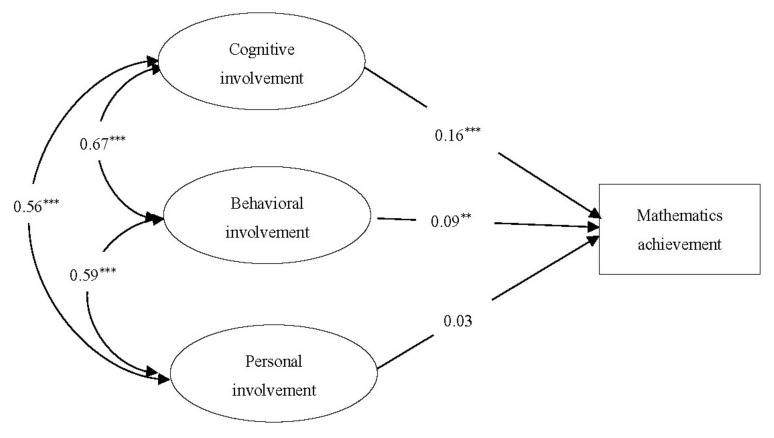
Path model depicting direct effort of parental involvement on mathematics achievement, controlling for gender, number of children at home, and socioeconomic status (SES). All coefficients shown are standardized. ** *p* < 0.01; *** *p* < 0.001.

**Figure 3 ijerph-18-09565-f003:**
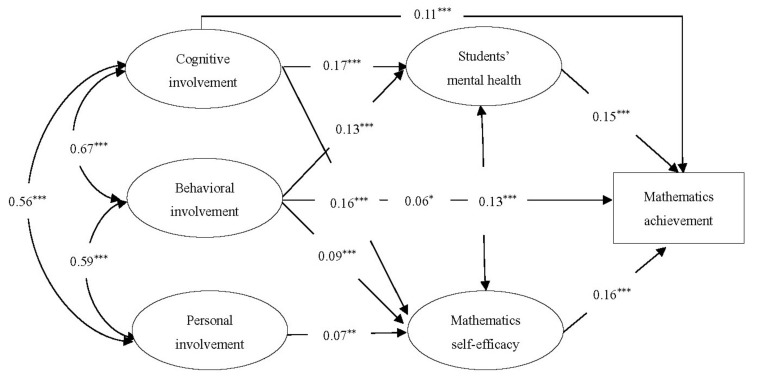
Final model depicting multiple mediation effects of students’ mental health and mathematics self-efficacy between parental involvement and mathematics achievement, controlling for gender, number of children at home, and SES. All coefficients shown are standardized and statistically significant. * *p* < 0.05; ** *p* < 0.01; *** *p* < 0.001.

**Table 1 ijerph-18-09565-t001:** Means (*M*), standard deviations (*SD*), and bivariate correlations of the study variables (*N* = 2866).

Variables	1	2	3	4	5	6	7	8	9
1. Cognitive involvement	1								
2. Behavioral involvement	0.73 ***	1							
3. Personal involvement	0.62 ***	0.71 ***	1						
4. Student mental health	0.26 ***	0.26 ***	0.23 ***	1					
5. Mathematics self-efficacy	0.22 ***	0.23 ***	0.20 ***	0.16 ***	1				
6. Mathematics achievement	0.27 ***	0.28 ***	0.24 ***	0.24 ***	0.25 ***	1			
7. Gender	−0.02	0.02	0.00	−0.03	−0.01	0.00	1		
8. Number of children at home	−0.07 ***	−0.10 ***	−0.10 ***	−0.03	−0.08 ***	−0.17 ***	0.02	1	
9. Socioeconomic status (SES)	0.11 ***	0.17 ***	0.11 ***	0.16 ***	0.12 ***	0.32 ***	0.00	−0.27 ***	1
*M*	4.15	4.22	4.28	4.51	3.31	84.08	0.47	0.47	0
*SD*	0.61	0.59	0.56	0.56	0.55	10.18	0.50	0.50	0.81

A 5-point Likert-type response scale was used for parental involvement and mental health. Self-efficacy could range from 1 to 4. Mathematics achievement scores could range from 0 to 100. *** *p* < 0.001.

**Table 2 ijerph-18-09565-t002:** Goodness-of-fit indices for the hypothesized model.

Goodness-of-Fit Indices	Hypothesized Model	Decision
*χ^2^/df*	1112.87 (*df* = 101, *p* = 0.00)	Rejected
*CFI*	0.93	Accepted
*TLI*	0.92	Accepted
*RMSEA* (90% CI)	0.06 (0.056, 0.062)	Accepted
*SRMR*	0.03	Accepted

Abbreviations: *χ^2^,* chi-squared value; *df*, degree of freedom; *CFI*, the comparative fit index; *TLI*, the Tucker–Lewis index; *RMSEA*, the root-mean-square error of approximation; *SRMR*, the standardized root-mean-square error; CI, confidence interval.

**Table 3 ijerph-18-09565-t003:** Standardized direct, indirect, and total effects for the final model from parental involvement to mathematics achievement.

Predictor and Covariate	Direct (95% CI)	Indirect (95% CI)	Total (95% CI)
**Predictor variable**			
Cognitive involvement	0.11 *** (0.07, 0.25)	0.06 *** (0.01, 0.17)	0.17 *** (0.06, 0.29)
Behavioral involvement	0.06 * (0.04, 0.08)	0.04 *** (0.02, 0.09)	0.10 ** (0.07, 0.27)
Personal involvement	0.01 (−0.03, 0.07)	0.02 ** (0.01, 0.10)	0.03 (−0.01, 0.04)
**Covariate**			
Gender	0.01 (−0.57, 0.55)	-	-
Numbers of child at home	−0.07 *** (−0.20, −0.05)	-	-
SES	0.24 *** (0.05, 0.60)	-	-

* *p* < 0.05; ** *p* < 0.01; *** *p* < 0.001.

## Data Availability

The dataset analyzed for the current report is not publicly available due to confidential information about the participants, but it may be requested from the corresponding author on reasonable request.

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
