# Peer review of "Relationship between Parental Involvement and Mathematics Achievement of Chinese Early Adolescents: Multiple Mediating Roles of Mental Health and Mathematics Self-Efficacy"

_ijerph, 2021, doi:10.3390/ijerph18189565_

Round 1
Reviewer 1 Report
The study “Relationship Between Parental Involvement and Mathematics 2 Achievement of Chinese Early Adolescents: Multiple 3 Mediating Roles of Mental Health and Mathematics 4 Self-efficacy” is generally fine. The introduction provides a background of the study and identify knowledge gaps, the methods are generally clear and suitable for the research hypotheses, results presentation is appropriate, and the discussion offers meaningful interpretation of the results. Below are some comments for the authors’ consideration.
Introduction:
- It is helpful to provide a little more rationale about why there is a need to specifically focus on math achievements.
- Provides a little more specifics about how the three dimensions of parental involvement contribute to students’ academic achievement differently
- Some statements may be not precise and should be refined. For example, the statement “In recent years, parental involvement has been a new focus of public concern, which was largely defined as ‘parents’ interaction with schools and with their children to 58 promote academic success” may be imprecise. Parental involvement has long been a research focus concerning various child wellbeing, and it is much more than on students’ academic wellbeing.
- The hypotheses can be clearer. For example, in the first hypothesis, parental involvement and two mentions of parental involvement were stated together. In this case, it is better to make it clear that the cognitive and behavioral involvement are two dimensions of the parental involvement. By the way, why only two not all three dimensions were mentioned?
Methods
- The SES is a combination of parental education and family income. Why not use them separately?
- A little more specifics about mathematics achievement measure, such as when it was taken.
The paper’s writing can be further polished.
Reviewer 2 Report
Dear authors,
Their study is quite interesting for the subject they have chosen, to see the relationship between parental involvement and academic performance in the subject of mathematics. However, you need to make some changes to improve it, which are the following:
- First, both in the confirmatory factor analyzes and in the models (of measurement and structural mediation), in my opinion, the RMSEA confidence interval is missing to complete the fit indices.They should include it. Furthermore, as a suggestion, they could include other internal consistency indices such as the McDonald's omega, the mean variance extracted or the composite reliability.
- On the other hand, I would need to know a little more how the SES variable was created, including the resulting categories.
- In the results part, I would like, if possible, to see the results of the independent samples t-tests.
- The measurement model, I see that it could be improved a little more, observing the adjustment indices.I suggest you try to improve it.
- Regarding indirect effects, I see that you have only included total indirect effects.They must break them down to know each one of them, in addition to including the confidence intervals of each of the effects, whether direct, indirect or total.
